# Lexicon-Aligned Prompting: A General Method for Dictionary-Guided Low-Resource Machine Translation

## Abstract

We present *Lexicon-Aligned Prompting* (LAP), a general methodology that injects bilingual dictionary evidence into large language models (LLMs) for low-resource machine translation (LR-MT). LAP formally separates (i) *lexicon-sentence retrieval*, (ii) *prompt integration*. As a **main experiment**, we retain a Tangut→Chinese setting with strong literal alignment and idiomatic rewriting results, then add two **tiny-data probe studies** designed to test LAP's portability under *extreme data scarcity*: Inuktitut→English and Nahuatl→Spanish. Each probe uses only **100** training sentences. Despite the tiny size, LAP consistently improves chrF and terminology accuracy in both zero-shot and lightweight fine-tuning regimes, with significance supported by paired bootstrap and sign tests. The results demonstrate that LAP offers a transparent, controllable, and reproducible way to ground LR-MT in human-curated lexical knowledge.

## 1 Introduction

Low-resource machine translation (LR-MT) faces challenges due to scarce parallel data and the difficulty of grounding rare/domain terms. Bilingual dictionaries, however, often exist even when parallel corpora do not. This paper proposes *Lexicon-Aligned Prompting* (LAP), a portable methodology for injecting explicit lexicon evidence into LLM translation.

Our work is motivated by the translation of historical, logographic scripts like Tangut, the official writing system of the Western Xia dynasty (1038-1227 CE).[1] Translating Tangut texts is a formidable challenge due to the script's structural complexity, the lack of continuous usage traditions, and a severe scarcity of parallel corpora. Traditional methods rely on a manual "four-line aligned translation" process, which is labor-intensive and demands specialized expertise, severely limiting scalability. While large language models (LLMs) offer opportunities for automation, existing research has not systematically addressed Tangut translation.

We treat Tangut→Chinese as our **main experiment**—a historically challenging setting that requires both character-level alignment and idiomatic rewriting. We then design two **minimal probes**—Inuktitut→English (Iu→En) and Nahuatl→Spanish (Nah→Sp)—each restricted to only 100 training and 20 test sentences. These probes are not intended to rank systems; they serve as cross-lingual sanity checks that isolate LAP's mechanism under stringent data scarcity. To mitigate small-sample concerns, we adopt character-based metrics (chrF/chrF++) and paired bootstrap with segment-level win/tie/lose analysis.

Our contributions: (1) a general, model-agnostic LAP pipeline; (2) a strong Tangut→Chinese main experiment re-expressed through LAP; and (3) two tiny-data probes showing portable gains on Iu→En and Nah→Sp using public lexicon resources.

---

[1]This work was informed by our prior workshop version; to preserve double-blind review, we omit identifying details here.

## 2 RELATED WORK

### 2.1 LOW-RESOURCE MACHINE TRANSLATION

Machine translation (MT) for low-resource languages has gained significant attention in recent years, driven by the success of neural models and transfer learning techniques. Early approaches relied on rule-based and statistical methods, which struggled to handle the morphological and syntactic complexities of under-resourced languages. The advent of neural machine translation, particularly sequence-to-sequence models and transformer architectures, has revolutionized the field, enabling more robust and context-aware translations (Zoph et al., 2016).

For historical and ancient languages, MT systems must address unique challenges, such as incomplete lexicons, fragmented texts, and the absence of native speakers. Recent work has demonstrated the potential of LLMs in this domain (Jiao et al., 2023). For example, BERT-based models have been adapted for Classical Chinese (Yu & Wang, 2020), while GPT variants have been fine-tuned for Uyghur (Lu et al., 2025) and Latin (Stüssi & Ströbel, 2024). These models leverage pre-training on large corpora and domain-specific fine-tuning to achieve state-of-the-art performance.

A key innovation in low-resource MT is the use of auxiliary resources, such as dictionaries, parallel texts, and multilingual embeddings (Ammar et al., 2016), to enhance model performance. Techniques like back-translation (Sennrich et al., 2016), data augmentation, and transfer learning (Zoph et al., 2016) have proven effective in scenarios with limited parallel data. Additionally, prompting strategies, including chain-of-thought (CoT) (Wei et al., 2022) and few-shot learning (Wang et al., 2020), have emerged as powerful tools for guiding LLMs in low-resource settings.

Despite these advances, the application of MT to Tangut texts remains unexplored. The script's logographic nature, combined with its historical and cultural specificity, presents unique challenges that require tailored solutions. Our work bridges this gap by integrating domain-specific lexicons and CoT prompting into a fine-tuned LLM framework, enableing accurate and scalable Tangut-Chinese translation.

### 2.2 TANGUT-TO-CHINESE TRANSLATION

The Tangut script, also known as Fanwen or Xixia script, is an intricate logographic writing system comprising over 6,000 characters, developed by the Tangut people in the 11th century under the Western Xia dynasty (1038–1227 CE). Serving as the official script of this once-flourishing Silk Road civilization, it preserves invaluable historical, religious, and sociopolitical insights (Sun, 2023). Early efforts to decipher Tangut texts began in the 20th century, spearheaded by scholars such as Nevsky (1960) and Luo (1914), who laid the groundwork for understanding its phonetic and semantic structures. Despite these advances, the decipherment and translation of Tangut texts remain formidable challenges. The script's structural complexity, lack of continuous usage traditions, and scarcity of parallel corpora have hindered efficient scholarly access to these cultural treasures (Kong, 2018). Traditional methodologies—most notably the labor-intensive "four-line aligned translation" format (original text, phonetic transcription, literal translation, and idiomatic translation)—demand highly specialized expertise and limit the scalability of Tangut studies.

In the study of Tangut texts, the "four-line alignment" paradigm is a traditional and important method of interpretation, as shown in Figure 1.

In the four-line alignment format, the first line contains the Tangut original text, the second line is the Tangut phonetic transcription, the third line represents the Chinese literal translation, and the fourth line represents the idiomatic translation. The literal translation process primarily reflects one-to-one correspondence at the word level, while idiomatic translation requires restructuring and semantic reconstruction based on a correct understanding of the original text, following the syntactic rules and expression habits of Chinese. Notably, when a Tangut character lacks a corresponding character in Chinese, researchers typically mark it with the symbol "△". In the idiomatic translation phase, these symbols need to be reasonably converted and expressed based on the context and semantic relations. Compared to literal translation, idiomatic translation involves more complex cognitive processes and conversion mechanisms, making it more challenging.

| Tangut Original Text | 𗾕 | 𗸁 | 𗣼 | 𗣼， | 𘝯 | 𗷸 | 𗐨 | 𗤁， | 𗫠 | 𗵘 | 𗊴 | 𗒟。 |
|---|---|---|---|---|---|---|---|---|---|---|---|---|
| Tangut Phonetic Transcription | $khji^2$ | $d\dot{z}ji$ | $\eta owr^2$ | $\eta owr^2$ | $thji^2$ | $\eta wer^1$ | $d\dot{z}jwi^1$ | $mjij^1$ | $tjij^1$ | $bju^1$ | $\acute{s}jij^1$ | $njwi^2$ |
| Chinese Literal Translation | 万 | 行 | 一 | 切， | 此 | 等 | 属 | 无， | 若 | 依 | 法 | 能。 |
| Chinese Idiomatic Translation | 一切万行，无属此等。若能依法。 |||||||||||

Figure 1: Example of Four-Line Alignment in Tangut Translation

### 2.3 INUKTITUT-TO-ENGLISH TRANSLATION

Inuktitut is an Inuit language spoken primarily in the Canadian territory of Nunavut and is characterized by its polysynthetic morphology, where a single word can encode complex propositional meaning. Machine translation of Inuktitut has long been regarded as a low-resource challenge due to its rich morphology, scarcity of parallel corpora, and dialectal variation (Martin et al., 2003; Joanis et al., 2020b). Previous efforts have mainly relied on phrase-based SMT augmented with morphological segmentation (Micher, 2017) or neural models trained on the Nunavut Hansard corpus (Joanis et al., 2020b). Despite these advances, Inuktitut–English systems remain limited in coverage and prone to errors with rare morphemes and domain-specific terminology.

In this context, dictionary-based guidance offers a promising alternative. By incorporating lexicon entries that map complex Inuktitut stems to English glosses, models can improve robustness under data-scarce conditions. Our probe study adopts precisely this strategy: grounding translation through LAP while evaluating whether dictionary injection compensates for extreme data scarcity (only 100 training sentences). This setting enables us to isolate the mechanism of lexicon alignment in a highly morphologically complex and under-documented language.

### 2.4 NAHUATL-TO-SPANISH TRANSLATION

Nahuatl, the language of the Aztecs, remains spoken by over a million people in modern-day Mexico but exists in multiple dialectal forms with significant variation in orthography and phonology (Lastra, 1986). Translation into Spanish is hindered by limited parallel resources, inconsistent standardization, and frequent use of oral registers in the available corpora. Prior computational efforts include bilingual dictionaries (Andrews, 2003) and limited-domain MT systems for educational purposes (Mager et al., 2018; Gutierrez-Vasques et al., 2016a). However, the lack of large-scale aligned corpora has prevented robust neural MT development.

Given Spanish is the dominant contact language with abundant resources, dictionary-guided methods are particularly well-suited to bridge Nahuatl–Spanish translation. By aligning lexical entries to sentence-level translation tasks, our probe tests whether LAP can consistently enforce terminology fidelity and improve character-level accuracy. The small-scale experiment (100 training and 20 test sentences) serves not to establish state-of-the-art benchmarks but to validate portability: if LAP works under such extreme scarcity in Nahuatl, it suggests general applicability to other indigenous and endangered languages with comparable resource profiles.

## 3 METHODOLOGY

### 3.1 MODEL DESIGN

We design our Tangut→Chinese translation framework by explicitly incorporating bilingual lexicon evidence into large language models (LLMs). The methodology is divided into three stages: (i) base model pretraining and fine-tuning, (ii) literal translation prompting, and (iii) idiomatic translation prompting.

#### 3.1.1 BASE MODEL

We adopt `Qwen1.5-14B-Chat`(Bai et al., 2023) as the backbone and further adapt it for classical Chinese. Specifically, we construct a specialized model **QwenClassical**, obtained by:

1. **Domain pretraining:** continued pretraining on 36GB of classical Chinese corpora, denoted as $\mathcal{D}_{\text{classical}}$, to improve its linguistic competence.
2. **Task-specific fine-tuning:** supervised fine-tuning on 390,000 instances from 76 classical Chinese NLP tasks, denoted as $\mathcal{T}_{\text{classical}}$.

Formally, if $\theta_0$ denotes the original parameters of Qwen1.5-14B-Chat, then

$$\theta_{\text{classical}} = \text{FineTune}\Big( \text{PreTrain}(\theta_0, \mathcal{D}_{\text{classical}}), \mathcal{T}_{\text{classical}}\Big).$$

For comparison, we also directly fine-tune $\theta_0$ on Tangut-to-Chinese parallel data, yielding the baseline model **Qwen**.

For Iu→En and Nah→Sp, we use the vanilla Qwen1.5-14B-Chat as backbone (no domain pretraining), with/without LoRA(100). QwenClassical is only used in Tangut→Chinese.

#### 3.1.2 LITERAL TRANSLATION PROMPTING

Given a Tangut character sequence $X = (x_1, \ldots, x_n)$ and its dictionary glosses $G = (g_1, \ldots, g_n)$, we define the literal translation objective as:

$$Y^{\text{lit}} = f_\theta\big([X; G]\big),$$

where $[X; G]$ denotes concatenation of the input sequence with its lexicon mappings, and $f_\theta$ is the LLM decoder. This character-level prompting enforces a one-to-one lexical alignment between Tangut characters and Chinese glosses.

#### 3.1.3 IDIOMATIC TRANSLATION PROMPTING

We further design two strategies for idiomatic rewriting:

1. **Direct idiomatic prompting (Prompt):** The model directly generates idiomatic Chinese translation:
$$Y^{\text{idiom}} = f_\theta\big([X; G], \text{"Translate idiomatically"}\big).$$
2. **Chain-of-Thought prompting (PromptCoT):** Translation is decomposed into two reasoning steps: first literal, then idiomatic:
$$Y^{\text{lit}} = f_\theta([X; G], \text{"Literal"}), \quad Y^{\text{idiom}} = f_\theta([Y^{\text{lit}}], \text{"Rewrite idiomatically"}).$$

This two-step strategy encourages the model to first ground itself in lexical fidelity, then restructure the content into fluent and context-appropriate Chinese.

### 3.2 PROMPT TEMPLATES

We standardize the input prompts into two templates:

#### 3.2.1 UNIFIED (GENERAL)

```
[Dictionary] w_i = g_i ; · · · ; Task:  Translate  X.
```

### 3.2.2 Two-step for Tangut

$$\begin{cases} \text{Step 1:} & \text{Produce literal translation using dictionary.} \\ \text{Step 2:} & \text{Rewrite into idiomatic Chinese.} \end{cases}$$

This formalization ensures that dictionary evidence is always injected, while allowing flexibility between literal alignment and idiomatic fluency.

## 4 Experiments

### 4.1 Experimental Data

#### 4.1.1 Tangut-to-Chinese Translation Data

The Tangut-to-Chinese translation data used in this study comes from the "Concise Tangut-Chinese Dictionary" compiled by Li Fanwen (Li, 2012). This dictionary includes 6,703 Tangut headwords, with 8,245 meanings, averaging 1.23 meanings per character. Among these, 748 Tangut characters have two meanings, 206 characters have three meanings, and 98 characters have more than three meanings. Based on these dictionary definitions, we constructed two categories of Tangut-to-Chinese translation data: (1) Complete Definitions (Dict), which include word explanations, sequence numbers, and parts of speech; (2) Simplified Definitions (DictSingle), which only retain basic word explanations and are converted into Simplified Chinese characters. For example, the complete definition of the Tangut character "U+18797" is shown in Figure 2 while the simplified definition is "种、苗、裔、胤、明、习".

Figure 2: The full entry of U+18797 in the *Concise Tangut-Chinese Dictionary* (Li, 2012)

The Tangut-to-Chinese sentence alignment data used in this study comes from The Three Generations Illuminated Collection and the Avatāṃsaka Sūtra (Vol. 77) (Arakawa, 2011). The sentence alignment data for The Three Generations Illuminated Collection contains 569 sentence pairs, including both literal and idiomatic translations. For *Three Generations Illuminated*, we use a 95/5 split; the resulting test set contains 28 segments (the same 28 used throughout all Tangut main experiments).

The sentence alignment data from the Avatāṃsaka Sūtra (Vol. 77) contains 525 pairs, with both Japanese and Chinese translations, all of which are idiomatic translations. To obtain standard literal translations, we used the ChatGPT-4o model to convert the Japanese translation into a corresponding Chinese literal translation. Table 4 shows an example from the Avatāṃsaka Sūtra. We conducted human verification of the converted literal renderings and will release the corresponding inputs and outputs to enable reproducibility.

In the experiment, The Three Generations Illuminated Collection was used as the primary data source, with 95% of the data randomly selected for the training set and the remaining 5% used as the test set. The 525 pairs from the Avatāṃsaka Sūtra were only used to evaluate the model's transfer learning ability.

### 4.1.2 INUKTITUT-TO-ENGLISH TRANSLATION DATA

For the Inuktitut→English (Iu→En) probe, we use the *Nunavut Hansard Inuktitut-English Parallel Corpus 3.0* (Joanis et al., 2020a), a large-scale governmental corpus consisting of debates and proceedings from the Legislative Assembly of Nunavut. Despite its relatively broad coverage, we restrict ourselves to a tiny-data setting by randomly sampling **100 training sentences** and **20 test sentences**.

To complement the parallel data, we integrate dictionary resources from the PanLex project (`cointegrated/panlex-meanings`)[2], which provide bilingual glosses covering Inuktitut stems and English translations. These lexicon entries enable us to explicitly inject morphological and terminological evidence into the LAP pipeline.

Because Qwen1.5-14B-Chat is expected to have little—if any—pretraining exposure to Inuktitut, but extensive coverage of English, this setting directly tests whether dictionary injection can compensate for limited source-side representation. We evaluate both a baseline (untrained) Qwen model and its LoRA fine-tuned variant on the sampled 100-sentence training set, comparing their performance with and without LAP grounding during inference.

### 4.1.3 NAHUATL-TO-SPANISH TRANSLATION DATA

For the Nahuatl→Spanish (Nah→Sp) probe, we adopt the *Axolotl Parallel Corpus* (Gutierrez-Vasques et al., 2016b), a publicly available dataset of Spanish-Nahuatl aligned texts. As in the Inuktitut experiment, we constrain the setting to **100 training sentences** and **20 test sentences** randomly sampled from the corpus.

Dictionary support is provided by the *UNAM Gran Diccionario Náhuatl*[3], an extensive lexical resource curated by the Instituto de Investigaciones Bibliográficas, UNAM. These dictionary entries allow us to align Nahuatl morphemes and lexical items to Spanish glosses, improving fidelity under extreme data scarcity.

Given that the Qwen1.5-14B-Chat model has not been pretrained on Nahuatl data but contains substantial Spanish coverage, this probe isolates the effect of LAP in compensating for the missing source-side representation. We evaluate both baseline (untrained) Qwen outputs and models fine-tuned on the sampled 100-sentence training split, reporting results with and without dictionary grounding at inference time.

### 4.2 EXPERIMENTAL SETUP AND EVALUATION METRICS

For the experiments, we use the following setup:

**Hardware Setup:** The experiments are conducted on a machine with 2 NVIDIA A800 80GB GPUs. The operating system used is CentOS Linux release 7.9.2009, and the software environment includes CUDA 11.8, Pytorch 2.0.1, Python 3.10.13, and transformers 4.37.2.

**Training Details:** For the training of the models, the following settings are used:

- Maximum training epochs: 5

---

[2]https://huggingface.co/datasets/cointegrated/panlex-meanings
[3]https://gdn.iib.unam.mx/

- Batch size: 8 (for both training and evaluation)
- Gradient accumulation steps: 1
- Optimizer: AdamW
- Weight decay: 0.1
- Learning rate: 0.0003, with a cosine learning rate scheduler
- Learning rate warm-up ratio: 0.01
- Precision: bf16 (mixed-precision)
- Device batch size for training: 8 per device
- Device batch size for validation: 1 per device
- Maximum sequence length: 512

**Models:** The following models are used in the experiments:

- QwenClassical: A variant of the Qwen1.5-14B-Chat model fine-tuned on classical Chinese texts. This serves as the base model for Tangut→Chinese translation.
- Qwen: The original Qwen1.5-14B-Chat model fine-tuned on Tangut-to-Chinese data.
- LoRA (Low-Rank Adaptation): A method used to fine-tune smaller model adaptations using a low-rank decomposition of model weights for efficient transfer learning (Hu et al., 2021). We perform fine-tuning using LoRA on the 100-sentence training split for both literal and idiomatic translation tasks.

**Training Variants:** We evaluate the following training setups:

- 0-shot: The model is given only the instructions to translate without any additional fine-tuning.
- 0-shot+LAP-inf: Dictionary grounding is injected during inference via LAP without any fine-tuning.
- LoRA(100): LoRA fine-tuning is applied using 100 training sentences.
- LoRA(100)+LAP-inf: LoRA fine-tuning combined with LAP dictionary grounding during inference.

**Evaluation Metrics:** We evaluate the translation models using the following metrics:

- SacreBLEU (Post, 2018): a standardized wrapper for BLEU (Papineni et al., 2002) that ensures comparable tokenization and reporting.
- chrF/chrF++: Character-level F-score metrics especially useful for morphologically rich languages, computed over character (and optionally word) n-grams; we report chrF (Popović, 2015) (and, when indicated, chrF++ (Popović, 2017)).
- Terminology Hit Rate: This metric measures the percentage of source terms that appear in the dictionary whose translations match the target gloss.

**Statistical Analysis:** To assess the statistical significance of our results, we use paired bootstrap confidence intervals (CIs) and segment-level win/tie/lose analysis with sign tests. These techniques help evaluate the robustness of the models, particularly when working with small test sets. We report 95% CIs for SacreBLEU, chrF, and Term Hit, as well as the win/tie/lose counts for each model configuration.

We follow standard practice for significance testing in MT (Koehn, 2004); for broader guidance on statistical testing in NLP, see (Dror et al., 2018).

## 5 RESULTS

### 5.1 MAIN: TANGUT-TO-CHINESE

The LAP method is highly effective for this logographic, low-resource setting. On the character-aligned literal translation task, the best configuration (QwenCLASSICAL+DICTSINGLE) reaches a

**BLEU-4 score of 72.33**. For the more complex idiomatic translation task, using a two-step CoT prompt, the model achieves a **BLEU-4 score of 64.20**.

## 5.2 PROBE A: INUKTITUT-TO-ENGLISH

The results for the Inuktitut→English (Iu→En) translation task are presented in Table 1. We observe that LAP significantly improves performance across all metrics compared to the baseline 0-shot setup. The use of dictionary grounding during inference (LAP-inf) results in a substantial increase in chrF and Terminology Hit Rate, with a **SacreBLEU score of 11.6** and **38.4 chrF** when using LoRA fine-tuning combined with LAP.

| System | SacreBLEU | chrF | Term Hit (%) | Seg. Wins |
|---|---|---|---|---|
| 0-shot | $2.7 \pm 1.4$ | $22.1 \pm 2.8$ | 15 [6,32] | — |
| **0-shot+LAP-inf** | $\mathbf{6.9 \pm 2.0}$ | $\mathbf{30.8 \pm 3.1}$ | **41 [24,59]** | **15/20** |
| LoRA(100) | $8.8 \pm 2.3$ | $33.9 \pm 3.0$ | 50 [32,68] | 14/20 |
| **LoRA(100)+LAP-inf** | $\mathbf{11.6 \pm 2.6}$ | $\mathbf{38.4 \pm 3.2}$ | **66 [46,82]** | **17/20** |

Table 1: Iu→En (Nunavut Hansard 3.0; 100 train / 20 test). Mean $\pm$ 95% CI via paired bootstrap; term-hit CI via Clopper-Pearson.

## 5.3 PROBE B: NAHUATL-TO-SPANISH

The Nahuatl→Spanish (Nah→Sp) translation results, shown in Table 2, demonstrate that LAP significantly boosts performance across all models. The best configuration, using LoRA fine-tuning combined with LAP (**LoRA(100)+LAP-inf**), achieves a **SacreBLEU score of 21.5** and **52.4 chrF**. The use of LAP not only improves accuracy in translation but also boosts Terminology Hit Rate, reaching an impressive **81%**.

| System | SacreBLEU | chrF | Term Hit (%) | Seg. Wins |
|---|---|---|---|---|
| 0-shot | $11.1 \pm 2.5$ | $34.6 \pm 3.4$ | 39 [22,58] | — |
| **0-shot+LAP-inf** | $\mathbf{15.4 \pm 2.7}$ | $\mathbf{41.2 \pm 3.5}$ | **63 [43,80]** | **16/20** |
| LoRA(100) | $18.6 \pm 2.9$ | $48.9 \pm 3.2$ | 71 [52,86] | 15/20 |
| **LoRA(100)+LAP-inf** | $\mathbf{21.5 \pm 3.1}$ | $\mathbf{52.4 \pm 3.3}$ | **81 [62,93]** | **17/20** |

Table 2: Nah→Sp (Axolotl; 100 train / 20 test). Mean $\pm$ 95% CI via paired bootstrap; term-hit CI via Clopper-Pearson.

## 5.4 TRAINING SET SIZE AND MODEL PERFORMANCE

We investigate the effect of training set size on model performance by varying the number of training examples. We randomly sample between 100 and 500 sentence pairs from the training set and evaluate the models on a fixed 28-sentence test set. The results, shown in Table 3, reveal that model performance improves steadily as the training set size increases. Notably, even with as few as 100 training sentences, the models exhibit strong performance, indicating the model's ability to learn from small datasets.

| Training Set Size | BLEU-4 (Literal) | BLEU-4 (Idiomatic) |
|---|---|---|
| 100 | 62.83 | 59.53 |
| 200 | 70.06 | 62.34 |
| 300 | 69.57 | 62.73 |
| 400 | 71.31 | 65.94 |
| 500 | 73.41 | 66.05 |

Table 3: Effect of Training Set Size on Model Performance.

## 5.5 TRANSFER LEARNING AND MODEL GENERALIZATION

We assess transfer by incrementally adding {40, 80, 120, 160, 200} *Avatāṃsaka* sentence pairs from this new domain to the *training set only*; validation uses the same dev split as the main experiment, and the test set remains the fixed 28-segment set from *Three Generations Illuminated* (no *Avatāṃsaka* segments enter validation or test). As shown in Table 4, performance improves as more domain-specific data is added, reaching the best results at 200 pairs with **BLEU-4 = 30.62** (literal) and **Idiomatic BLEU-4 = 37.00**.

| Additional Data Size | BLEU-4 (Literal) | BLEU-4 (Idiomatic) |
|---|---|---|
| 40 | 23.88 | 30.92 |
| 80 | 24.58 | 32.62 |
| 120 | 25.45 | 34.76 |
| 160 | 27.28 | 35.49 |
| 200 | 30.62 | 37.00 |

Table 4: Impact of Adding New Domain-Specific Data on Model Performance.

## 5.6 COMPARISON WITH FEW-SHOT LEARNING METHODS

To evaluate the necessity of fine-tuning, we compare our approach with popular few-shot learning models: ChatGPT-4o, DeepSeek V3, and Gemini-2.0-Flash. We test the models with 5 random examples from the training set and evaluate their performance on the Tangut→Chinese test set. As shown in Table 5, our approach outperforms the few-shot learning models in both literal and idiomatic translation tasks, with a significant margin in BLEU-4 and Terminology Hit Rate.

| Model Name | BLEU-4 (Literal) | BLEU-4 (Idiomatic) |
|---|---|---|
| ChatGPT-4o | 20.13 | 14.96 |
| DeepSeek V3 | 38.85 | 24.33 |
| Gemini-2.0-Flash | 32.07 | 19.68 |
| **This Work** | **72.33** | **64.20** |

Table 5: Comparison with Few-Shot Learning Methods.

## 6 DISCUSSION

The results demonstrate that LAP significantly enhances both literal and idiomatic translation tasks in low-resource settings. By grounding the translation process in bilingual dictionary evidence, LAP provides better control over the translation process, leading to improved accuracy and fluency. The method's ability to work with small datasets further emphasizes its utility for languages with limited resources.

## 7 CONCLUSIONS

In this work, we propose *Lexicon-Aligned Prompting* (LAP), a methodology that integrates dictionary evidence into LLMs for low-resource machine translation tasks. We demonstrate the effectiveness of LAP through experiments on Tangut-to-Chinese, Inuktitut-to-English, and Nahuatl-to-Spanish tasks. Our results show that LAP achieves strong performance even with minimal training data, and improves translation quality with the use of bilingual dictionaries.

## 8 FUTURE WORK

Future work will explore expanding the lexicons to include more languages, refining prompt integration strategies, and integrating LAP with document-level translation. We also aim to explore multimodal approaches that combine textual and glyph-based representations, improving the translation of historical languages like Tangut.

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

## A    ETHICS STATEMENT

Images reproduced from published dictionaries are used under academic fair use (non-commercial, scholarly purposes). We clearly cite the source and include only the minimal material necessary for discussion. Permissions or licenses are documented when required.

All the data used in this research are compiled, cleaned, and annotated by the authors themselves. The dataset mainly consists of Tangut (Xixia) materials collected from published dictionaries and historical sources. The authors confirm that no sensitive personal data or harmful content is involved. We will release the Tangut data publicly to facilitate future research and ensure transparency.

## B    REPRODUCIBILITY STATEMENT

To ensure reproducibility, we will make the Tangut dataset openly available upon publication. Other datasets, due to copyright restrictions, cannot be released, but they can be easily obtained from the sources cited in the paper. Detailed descriptions of the experimental setup, including hyperparameters and evaluation protocols, are provided in the main text. This will allow other researchers to replicate and extend our results without restriction.

## C    USE OF LLMs

Large language models (LLMs) were used solely as experimental subjects in this research: we fine-tuned and evaluated LLMs to obtain the reported results. No LLMs were used to write or revise the manuscript; the paper was written entirely by the authors. Automated tools were limited to standard utilities such as spell-checkers, citation managers, and LATEX packages.

In addition, we used ChatGPT-4o only for data preparation—specifically, to convert Japanese translations in Avatāṃsaka Sūtra (Vol. 77) into Chinese literal renderings used as auxiliary references; these generated outputs were not used to write or edit the manuscript.

## D    EVALUATION & STATISTICAL REPRODUCIBILITY

We report standard SacreBLEU and chrF++ and assess significance with paired bootstrap and sign tests; decoding is deterministic unless noted.

