# OpenReview forum: "Lexicon-Aligned Prompting: A General Method for Dictionary-Guided Low-Resource Machine Translation"
_ICLR.cc/2026/Conference — ICLR 2026 Conference Withdrawn Submission_

### Official Review · Reviewer_tARY · 2025-10-22

**Soundness:** 2
**Presentation:** 2
**Contribution:** 1
**Rating:** 2
**Confidence:** 3

**Summary:**

The paper proposes Lexicon-Aligned Prompting for low-resource machine translation by injecting bilingual dictionary evidence into prompts. It tackles Tangut to Chinese with a two-stage pipeline that first continues pretraining a Qwen1.5-14B-Chat variant on Classical Chinese, then applies task-specific fine-tuning, followed by literal-to-idiomatic prompting with an optional chain-of-thought step. Two tiny-data probes on Inuktitut to English and Nahuatl to Spanish, each with 100 training and 20 test sentences, show consistent gains.

**Strengths:**

Practical utility for engineering: The approach is straightforward to implement and likely to work well in real systems, making it appealing from an engineering perspective.

**Weaknesses:**

1. Methodological novelty is limited: The core technical ingredients—dictionary/terminology injection, and fine-tuning—are each well-explored. The contribution largely combines and migrates these elements to extreme low-resource settings. Such migration to harder/smaller data conditions does not, by itself, constitute ICLR-level methodological innovation. Worse still, the paper fails to discuss its methodological distinctions from any recent related work, which is a critical omission that urgently needs to be addressed. The “Future Work” items (e.g., document-level translation, multimodal) would materially strengthen the research novelty; consider bringing at least one of these into the current paper as an actual technical contribution rather than leaving it as future work.

2. Baselines are insufficient: The paper mainly compares against few-shot prompting of other models. This is not enough: there are many existing works that inject dictionaries/terminologies at training or decoding time. The differences in method and effectiveness relative to those lines are not made clear. Please add:

3. Experimental setup issues: It is unclear why BLEU-4 is reported instead of BLEU. In addition, the last row of Table 5 does not appear to have a corresponding setting or result elsewhere in the paper; please reconcile or correct this inconsistency.

**Questions:**

See Weaknesses.

---

### Official Review · Reviewer_5Rn2 · 2025-10-26

**Soundness:** 2
**Presentation:** 3
**Contribution:** 2
**Rating:** 2
**Confidence:** 4

**Summary:**

The paper proposes injecting lexical translation in LLM prompts to improve low-resource machine translation, and focuses on Tangut-Chinese, Inuktitut-English and Nahuatl-Spanish language pairs. For Tangut-Chinese, Qwen 14B was further pretrained with a Classical Chinese corpus, then finetuned on specific tasks also in Classical Chinese. To prompt for translation, dictionary glosses are concatenated with the input sequence. Idiomatic prompting involves two strategies: i) direct instruction to translate idiomatically given the input and glosses, ii) a two-stage pipeline where model first translates literally and rewrite the literal translation for more fluent and idiomatic writing. The authors also organized and re-translated part of the linguistic resources in Tangut to make them accessible. For other language pairs, finetuning is performed on 100 parallel sentences, before evaluating their performance on lexicon injection. The approach is effective against baseline.

**Strengths:**

The paper explored new language pairs in under-resourced translations. There is also clever adaptation of various Tangut resources, e.g.,  translating from Japanese to Chinese in Tangut-Japanese parallel corpus.

**Weaknesses:**

1. I'm struggling to see any novelty in the work. Dictionary-based / phrase-level prompting is not new in the LLM translation literature, see for example,

- https://arxiv.org/pdf/2302.07856
- https://aclanthology.org/2023.sigmorphon-1.2.pdf
- https://aclanthology.org/2024.findings-acl.152.pdf

Multi-stage translation with prompting: https://arxiv.org/abs/2409.06790, etc.

2. While the processed data might be valuable to low-resource NLP research, the authors are not the primary contributors in creating these resources.

3. Having 28 + 525 test instances for Tangut-Chinese is reasonable. However, evidence is lacking on other language pairs with only 20 test instances.

4. I suspect the authors also misunderstood transfer learning in Section 5.5. It is a well known fact that including training instances from different domains adds robustness and improves performance. The evidence for transfer learning here is weak at best.

**Questions:**

1. Apart from applying existing LLM MT methods on new language pairs, what are the fundamental contributions here?
2. Why not a larger test set for the other language pairs? Under low-resource setting, it is reasonable for test set to be larger than training set.
3. What do you mean by transfer learning in Section 5.5?

---

### Official Review · Reviewer_gmyp · 2025-10-31

**Soundness:** 3
**Presentation:** 2
**Contribution:** 2
**Rating:** 2
**Confidence:** 5

**Summary:**

This paper applies methods for extremely low resource translation with LLMs to the Tangut language. The adapted model performs better than zero-shot LLMs. There isn't enough novelty/contribution over closely related prior works, which are also barely discussed in the paper.

**Strengths:**

Tangut seems like an interesting domain, and there is an improvement over baselines. I'm not familiar with prior work on Tangut, but I can't really search to confirm because it would probably deanonymize the paper.

Results are also validated through probes on 2 additional languages.

**Weaknesses:**

The related work section is really lacking, specifically on works related to extremely low resource translation in LLMs (especially with in-context learning and reasoning e.g. using grammar notes or glosses) from the last 2-3 years. e.g. https://arxiv.org/abs/2302.07856 https://arxiv.org/abs/2309.16575 https://aclanthology.org/2024.findings-acl.925/ https://arxiv.org/abs/2410.18702 https://arxiv.org/abs/2502.11862 https://arxiv.org/abs/2402.19167v1 and many, many more

i.e., The paper's originality / generalizable takeaways are too weak. Or at the very least, if there is some unique contribution vs. these other works that I'm not seeing, it isn't expressed clearly in relation to them. I also don't see any particular new insights about these methods, e.g. new analysis of failure modes, scaling laws, etc.

**Questions:**

How does your work differ from the many prior works linked above?

Maybe I'm missing it, but what dataset is "Tclassical"? Appendix B says "Other datasets, due to copyright restrictions, cannot be released, but they can be easily obtained from the sources cited in the paper." but I don't see any source.

---

### Official Review · Reviewer_L5EN · 2025-11-01

**Soundness:** 2
**Presentation:** 2
**Contribution:** 2
**Rating:** 4
**Confidence:** 3

**Summary:**

This paper proposes Lexicon-Aligned Prompting (LAP) that is a method to inject bilingual dictionary knowledge into large language models (LLMs) to improve low-resource machine translation (LR-MT). LAP separates lexicon-sentence retrieval from prompt integration and demonstrates its effectiveness on Tangut -> Chinese, Inuktitut->English, and Nahuatl->Spanish translation tasks. Experiments show that LAP improves both literal and idiomatic translation quality, even under extreme data scarcity (as few as 100 training sentences). The method is transparent, controllable, and reproducible, providing a promising approach for LR-MT grounded in human-curated lexical knowledge.

**Strengths:**

- The experiments focus on extreme low-resource settings, reflecting practical scenarios for endangered or underrepresented languages.
- The proposed approach demonstrates improvements even with only 100 training sentences, highlighting its effectiveness.

**Weaknesses:**

- Current experiments are conducted on only three language pairs. Including additional low-resource language pairs would strengthen claims about generalizability.
- t is unclear how LAP performs when word-by-word or strong lexical alignment between source and target languages is weak or absent, which could limit applicability.

**Questions:**

- Can you test the proposed approach on additional low-resource language pairs to demonstrate generalizability?
- How sensitive is LAP to languages with weak or non-literal word-by-word alignment?
- In Figure 2, non-English characters are used without English translations, which may reduce accessibility for readers unfamiliar with the scripts. Adding corresponding English translations would be great.

---

### Note · Authors · 2025-11-20

I have read and agree with the venue's withdrawal policy on behalf of myself and my co-authors.